

# HERVs, immunity, and autoimmunity: understanding the connection

Matthew Greenig

Department of Life Sciences, Imperial College London, London, United Kingdom

## ABSTRACT

Since their discovery in the 1960s, further investigation into endogenous retroviruses (ERVs) has challenged the conventional view of viral sequences as exclusively parasitic elements. Once presumed to be a group of passive genetic relics, it is becoming increasingly clear that this view of ERVs, while generally accurate, is incorrect in many specific cases. Research has identified ERV genes that appear to be co-opted by their mammalian hosts, but the biological function of ERV elements in humans remains a controversial subject. One area that has attracted some attention in this domain is the role of co-opted ERV elements in mammalian immune systems. The relationship between ERVs and human autoimmune diseases has also been investigated, but has historically been treated as a separate topic. This review will summarize the current evidence concerning the phenotypic significance of ERVs, both in the healthy immune system and in manifestations of autoimmunity. Furthermore, it will evaluate the relationship between these fields of study, and propose previously-unexplored molecular mechanisms through which human endogenous retroviruses might contribute to certain autoimmune pathologies. Investigation into these novel mechanisms could further our understanding of the molecular basis of autoimmune disease, and may one day provide new targets for treatment.

# INTRODUCTION

It has long been known that up to 8% of the human genome is derived from a large number of viral elements (*International Human Genome Sequencing Consortium, 2001*; *Patel, Emerman & Malik, 2011*). These endogenous viral elements (EVEs) are present in modern humans as the result of ancient viral integration events into the germline cells of our ancestors. The process of viral germline integration is known as endogenization, and has likely occurred many times throughout mammalian evolution, with certain EVEs estimated to have integrated over 100 million years ago (*Lee et al., 2013*). Investigation into these ancient viral elements has given rise to a new field of study known as paleovirology, an area that focuses particularly on the evolutionary history and present-day biological significance of EVEs.

Predictably, the majority of EVEs appear to be derived from retroviruses (*Katzourakis & Gifford, 2010*), the only animal viruses that undergo genomic integration as part of their normal life cycle. Exogenous retroviruses are transmitted as (+) strand RNA, packaged

Corresponding author
Matthew Greenig, mg1316@ic.ac.uk

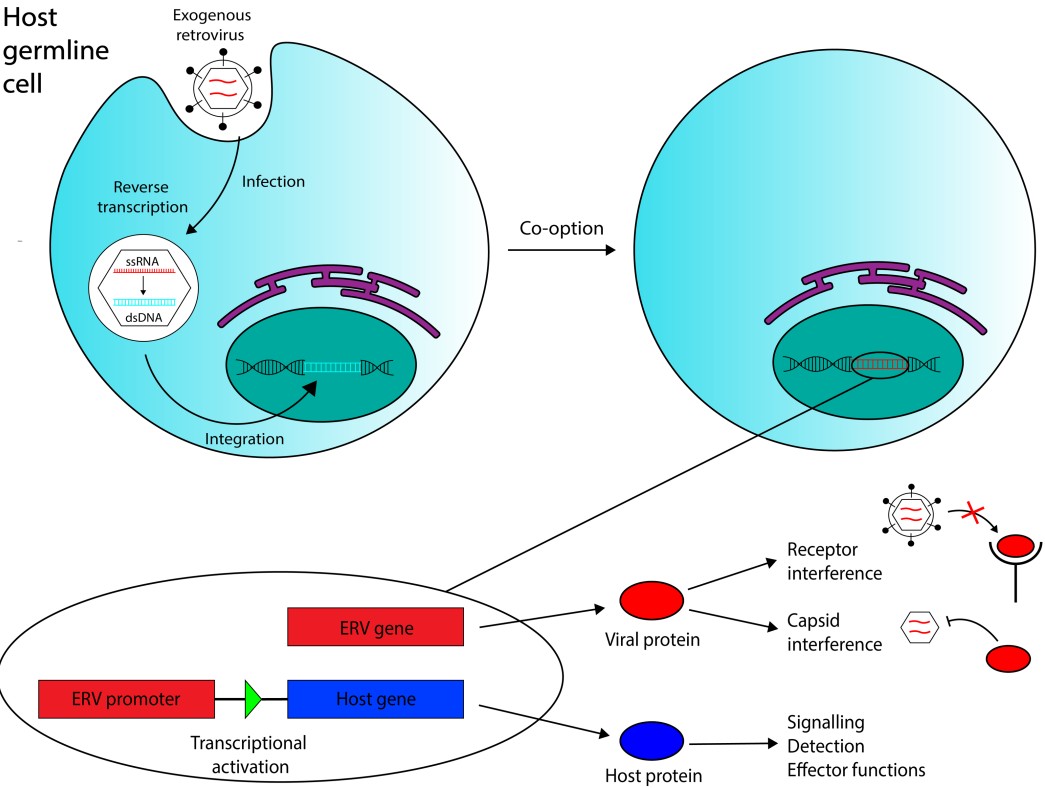

**Figure 1  Overview of the ERV co-option process with immune system functions annotated.** Retroviral endogenization occurs when an exogenous retrovirus integrates into the genome of a host germline cell. The integrated element can be co-opted to act as a coding gene or a promoter region for a host gene. Three functions of co-opted ERVs in the immune system that are discussed in this review are annotated. Recent evidence has also suggested that ERV RNA transcripts, in addition to ERV-encoded proteins and ERV-derived regulatory regions, may also mediate antiviral immunity in vertebrates.

in a protein capsid and lipid envelope. Upon entering the cell, they undergo a unique step in their life cycle that involves reverse transcription of their single-stranded RNA to form double-stranded DNA (dsDNA). The virus is then transported to the nucleus, where it mediates integration of the newly-formed viral dsDNA into the host's genome. This mechanism is displayed in Fig. 1. Retroviral endogenization mechanisms are well-characterized, and currently, ERVs constitute the most thoroughly investigated subset of EVEs.

Exogenous viruses are transmitted horizontally, moving between individuals whose cells provide the machinery required for the virus to replicate itself and release infectious particles, so that other susceptible individuals can be infected. Retroviruses can be transmitted in this way, but their permanent integration into the host cell's genome also allows for another mode of transmission to occur. If a retrovirus integrates into the genome of a germline cell that contributes to the formation of a nascent individual, the integrated provirus will be transferred from parent to offspring—a process known as vertical transmission. If retroviral integration occurs before the first cell division of a

zygote, all of the progeny's somatic cells, and at least half of their gametes, will contain the integrated retrovirus. The retroviral DNA is then present in and can be transferred through the organisms's germline; the virus has become endogenized. Immediately after endogenization, most new ERVs probably remain replication-competent, so viral re-integration events following endogenization are still possible; these would increase provirus copy number in the host and could increase the ERV's frequency in the gamete population, if the virus was transmitted between gametes. Indeed, some work has identified human ERV (HERV) families with particularly high copy numbers (200+ copies) (*Belshaw et al., 2005*). Yet, although we have identified actively-expressed ERVs in many other vertebrates (*Aswad & Katzourakis, 2012*), nearly all human ERVs discovered thus far have accumulated mutations rendering them replication-incompetent, and their protein products defective (*Weiss, 2016*). Most HERVs are not even actively transcribed (*Oja et al., 2007*). However, these generalizations do not apply to the entire group.

Some of the earliest evidence that certain HERVs might be expressed in modern humans came from the discoveries of Syncytins 1 and 2, two proteins that are expressed in developing human embryos. These proteins are full-length products derived from integrated retroviral envelope genes (*env*); they mediate fusion between placental cells, a process that is disrupted in their absence (*Mi et al., 2000*). As some of the earliest discovered examples of co-opted retroviral elements, syncytins have prompted further research into host domestication of endogenous retroviral genes. Later bioinformatics-based investigations revealed that a small but noteworthy proportion of ERVs in the human genome contain ORFs encoding for intact viral polyproteins (*Villesen et al., 2004*). The traditional view of ERVs assumes them to be defunct copies of ancient exogenous retroviruses, whose sequences have persisted in the human germline precisely because of accumulated mutations that prevent the virus from replicating, converting the ERV into a neutral locus. Neutral loci are expected to undergo mutational decay over large periods of time, so the presence of intact HERV ORFs in the human genome could be an indicator of relatively recent endogenization events. However, an alternative explanation suggests that these sequences have been selectively maintained throughout evolutionary history, after being co-opted for some beneficial function. These hypotheses are not mutually exclusive, and both would imply that certain HERV elements might still be actively expressed in modern day humans. One microarray-based assay observed HERV transcription in every sample of 19 different human tissue types (*Seifarth et al., 2004*), indicating that HERV expression might be relatively ubiquitous throughout the human body.

Taken together, these results suggest that the view of HERVs as inactive, neutral sequences does not apply to all endogenous retroviral elements in the human genome. It is well documented that ERVs are not only expressed, but produce phenotypic consequences in other vertebrates (*Aswad & Katzourakis, 2012*). Investigation into the function of co-opted ERVs in the immune systems of various vertebrate species has revealed complex, intimate relationships between endogenous retroviruses and their hosts (*Aswad & Katzourakis, 2012*; *Frank & Feschotte, 2017*). Despite the ubiquity of such relationships, the phenotypic effects of HERV elements on the immune systems of their human hosts largely remain a mystery. Accordingly, much controversy has emerged regarding the link between

HERVs and human autoimmune diseases. Increased HERV activity has been identified as a key feature in various forms of autoimmunity, and while mechanistic theories have been formulated to explain their contribution to disease pathogenesis (*Balada, Ordi-Ros & Vilardell-Tarrés, 2009*), there is still no consensus regarding the connection between HERVs and autoimmunity. Furthermore, the relationship between HERV function in the healthy immune system and dysfunction in autoimmune disease remains mostly unexplored. In the present paper, I review some of the existing evidence on interactions between endogenous retroviruses and mammalian immune systems, both in the context of normal immunity and autoimmunity. In addition, I utilize a unique cross-disciplinary framework for evaluating the relationship between HERV elements and human autoimmune diseases, incorporating principles from retrovirology, immunology, and our knowledge of ERV/host interplay in vertebrate immune systems. Using this cross-disciplinary approach, I formulate and present multiple novel hypotheses concerning HERV-mediated mechanisms of autoimmunity.

## Survey methodology

This review discusses a subject that exists at the interface between evolutionary biology, virology, and immunology. Therefore, sources from a wide range of journals were used, including those dedicated to specific research in either virology or immunology, as well as work published in journals with broader research themes, e.g., *Nature*. The vast majority of citations derive from primary research, and quantitative evidence is cited whenever available.

### EVE-Derived Immunity

Host-virus interactions over large evolutionary time scales are generally described using an 'arms race' model, in which the virus and host 'race' to evolve novelties that allow them to successfully outcompete one another. Such evolutionary competition appears to have driven the evolution of animal immune system genes to a huge extent (*Daugherty & Malik, 2012*). In the prototypical model for natural selection, evolutionary novelties develop through mutation of pre-existing genes. Viruses clearly have the edge in this regard, exhibiting higher mutation rates, faster generation times, and an astounding ability to handle mutational load (*Lauring, Frydman & Andino, 2013*). However, co-option of an endogenous viral gene provides a powerful weapon for their competing animal hosts, introducing an entirely novel element to the host-virus arms race, to which a competing virus has not developed resistance. Therefore, although retroviral integration can lead to disease, insertional oncogenesis, or disruption of host genes, co-opted ERVs can also offer a crucial edge to their hosts in the molecular arms race. This advantage might explain the observed ubiquity of ERV-encoded products in vertebrate immune systems (*Aswad & Katzourakis, 2012*; *Frank & Feschotte, 2017*), despite the improbability of ERV co-option events and the fitness costs associated with retroviral endogenization events.

Co-opted ERVs can be broadly divided into two categories: elements whose function derives from their encoded product(s), and elements that act as regulatory sequences in the host genome. Examples of both appear in mammalian immune systems (see Fig. 1).

While recent investigation has suggested that some EVEs confer antiviral immunity through non-coding RNAs (*Honda & Tomonaga, 2016*), most research to date regarding

ERV products has focused on ERV-encoded proteins. Restriction factors are a class of cellular proteins that actively inhibit viral replication (*Goff, 2014*), and certain co-opted ERVs have been identified to act as restriction factors against exogenous retroviruses. So far, no human ERVs with these properties have been discovered (*Frank & Feschotte, 2017*), but other research has identified a number of ERV-derived restriction factors in the murine model, as well as in sheep, chickens, and cats (*Aswad & Katzourakis, 2012*).

### Co-opted ERVs in the immune system

Fv-4 is one ERV-derived restriction factor that has been identified in mice. The *Fv-4* locus contains a fully intact *env* gene with >70% sequence similarity to *env* in multiple strains of exogenous murine leukemia virus (MuLV) (*Ikeda et al., 1985*). Fv-4 expression was later demonstrated to directly confer resistance to ecotropic MuLV in transgenic mice; host resistance to the virus also correlated with the levels of Fv-4 expressed (*Limjoco et al., 1993*). Further work has been done to elucidate the ERV-encoded protein's mechanism of action. While viral Env proteins usually bind to receptors on the cell surface to mediate cell entry, exogenous MuLV Env has been shown to be capable of interacting with its receptors intracellularly in the endoplasmic reticulum, as demonstrated by pulse-chase labelling (*Kim & Cunningham, 1993*). Kim and colleagues observed that this intracellular binding interfered with receptor maturation by inhibiting N-linked glycosylation of two residues, and that receptor binding to exogenous MuLV was significantly reduced in cells expressing MuLV Env. However, they also observed that mutant non-glycosylated forms of the receptor bind extracellular MuLV at the same efficiency as the wild type receptor, in the absence of intracellular MuLV Env. Thus, the ability of intracellular MuLV Env to inhibit MuLV infection could rely on intracellular retention of the MuLV receptor due to ligand binding, or, on competitive inhibition of virus binding sites, reducing the number of receptors available to exogenous MuLV on the cell surface. Figure 2 provides a general schematic for these resistance mechanisms. Later work also revealed that one of the amino acid residues at which Fv-4 differs from exogenous MuLV Env is necessary for the Env protein's fusogenic capabilities; exogenous Env that is modified with the Fv-4 mutation at this residue is unable to fuse with target cell membranes, and forms uninfectious virion particles (*Masuda & Yoshikura, 1990*; *Taylor, Gao & Sanders, 2001*). Due to its homology to exogenous MuLV Env, Fv-4 can likely be incorporated into MuLV virions, thereby producing uninfectious particles with non-fusogenic Env proteins. Thus, although its restriction mechanism has not been confirmed, Fv-4's antiviral activity can likely be attributed to its homology to MuLV Env. This example illustrates the broader point that certain ERVs have maintained high levels of structural similarity to their exogenous counterparts throughout evolutionary history, which could allow them to interact with host proteins that normally interact with exogenous viruses.

Non-Env restriction factors derived from ERVs have also been identified in the murine model. Fv-1 is one such protein, first observed in the 1970s to confer resistance to MuLV (*Pincus, Hartley & Rowe, 1971*). Fv-1 appears to be derived from a retroviral *gag* gene (*Best et al., 1996*), the domain responsible for encoding viral capsid elements. Fv-1 appears to restrict MuLV by a mechanism similar to that of the TRIM5 $\alpha$ restriction factor expressed

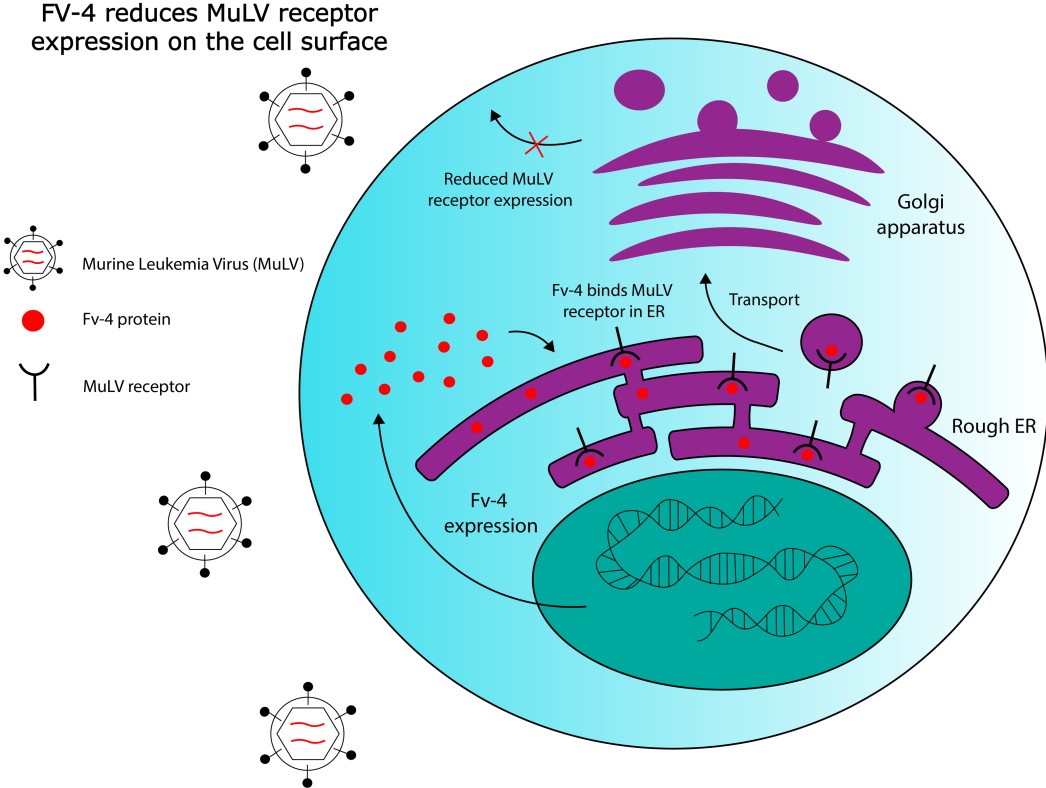

**Figure 2** **Hypothesized Fv-4 restriction mechanism in mice.** Fv-4 binds to MuLV receptors in the rough endoplasmic reticulum, and reduces their expression on the cell surface. Fv-4 can also likely be incorporated into MuLV virion particles in infected cells, reducing the number of infectious particles that are produced.

in humans, which also restricts MuLV (*Yap et al., 2004*). TRIM5 $\alpha$ is known to restrict retroviral infection through interaction with the viral capsid (*Sastri & Campbell, 2011*), and restriction by both Fv-1 and TRIM5 $\alpha$ relies on specificity at the same single amino acid in MuLV's capsid protein, suggesting a common mechanism of action (*Kozak & Chakraborti, 1996*; *Yap et al., 2004*). In addition, phylogenetic analysis of Fv-1 genes in different mouse subgenera has provided evidence of positive selection in different variants of Fv-1 (*Yan et al., 2008*). Specifically, dN/dS >1 was noted at three codons in a segment known to be involved in capsid interactions, indicating an evolutionary history of host-virus interactions at this site. Being derived from a retroviral *gag* gene, Fv-1's homology to retroviral capsid elements likely enables it to bind exogenous viral capsids, which themselves are assembled through high-affinity interactions between structurally identical monomeric subunits. Importantly, both Fv-4 and Fv-1 illustrate a broader theme of molecular mimicry in co-opted ERV elements: products with homology to components of exogenous viruses.

A separate area of research has investigated the role of co-opted HERVs as regulatory sequences, specifically retroviral long terminal repeat domains (LTRs). In exogenous retroviruses, LTRs function primarily as cis-regulatory sequences for viral genes (*Temin, 1981*). However, multiple examples of LTR co-option in mammals have been discovered
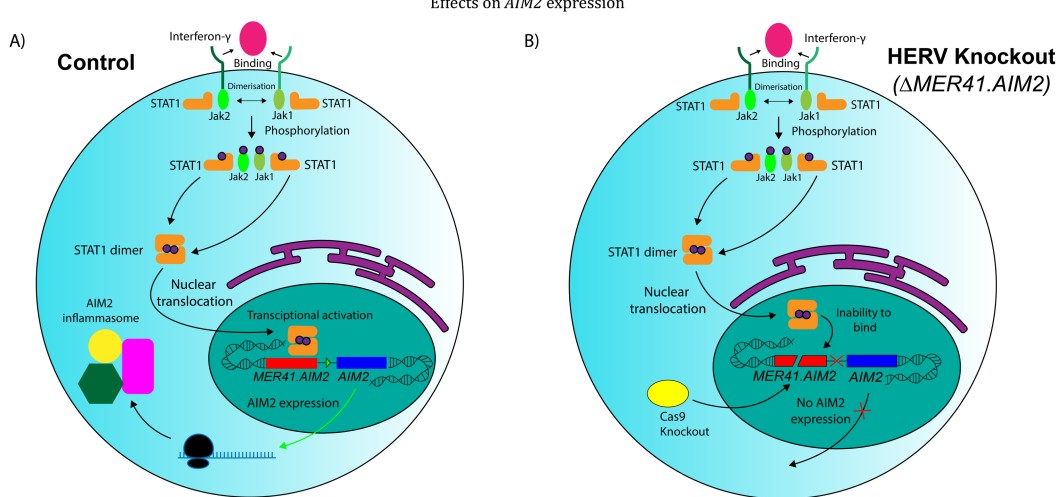

**HERV LTR knockouts in HeLA cells**
Effects on *AIM2* expression

**Figure 3  Effects of CRISPR/Cas knockout of a HERV regulatory element in the human immune system (adapted from (***Chuong, Elde & Feschotte, 2016***)**  Evidence obtained by *Chuong, Elde & Feschotte (2016)* suggests that CRISPR/Cas knockout of *MER41.AIM2* prevents expression of *AIM2* upon treatment with IFN-γ by inhibiting STAT1's binding to the promoter. These results imply a crucial role for the HERV element in cytokine signalling. (A) WT HeLa cells expressing *AIM2* when exposed to IFN-γ. (B) *MER41.AIM2*-knockout HeLa cells that do not express *AIM2* upon IFN-γ stimulation.

in recent years (*Franke et al., 2017*). These findings extend to immunology, where co-opted LTRs have been implicated in the human interferon (IFN) response (*Chuong, Elde & Feschotte, 2016*). Chuong and colleagues analysed the role of ERV LTRs in promoting the transcription of IFN-γ-stimulated genes. Their findings revealed that certain ERV LTRs are bound by STAT1 and/or IRF1, transcription factors that regulate interferon-induced genes. These LTRs were strongly enriched near interferon-stimulated sequences, suggesting a possible role for them in IFN- γ gene regulation. Chuong's group further investigated MER41, a family of HERVs shown to be a source of IFN-inducible binding sites. An LTR belonging to the MER41 family (*MER41.AIM2*) was shown to be the only STAT1 binding site within 50 kb of the IFN-γ-inducible gene *Absent in Melanoma 2* (*AIM2*) gene, a foreign DNA sensor involved in the innate immune response. *MER41.AIM2* thus appears to be a primary STAT1 promoter for *AIM2*, and CRISPR-Cas9 deletions in HeLa cell lines demonstrated that *MER41.AIM2* knockout rendered cells unable to express AIM2 upon IFN- γ treatment (*Chuong, Elde & Feschotte, 2016*), as shown in Fig. 3.

*Chuong, Elde & Feschotte (2016)* also found *MER41* LTR elements within 20 kb of three other interferon-inducible genes, and genetic knockouts showed that each of these ERV promoters contributed to expression of IFN- γ-induced genes. Interestingly, it was discovered separately that exogenous HIV-1 LTRs contain an IRF1 binding site (*Sgarbanti et al., 2008*), a feature that likely evolved as a way for the virus to regulate its expression in response to host immune signalling. When considered with *MER41*'s role in immunoregulation, these results imply that the evolution of host transcription factor

binding sites in exogenous viral sequences might actually facilitate co-option of ERV regulatory elements. Exogenous retroviruses that contain regulatory sequences with such binding sites might not require significant changes in nucleotide sequence to be used as host promoters, as they interact with host transcription factors in their exogenous form. Furthermore, because many retroviruses have evolved to infect immune system cells (*Coffin, Hughes & Varmus, 1997*), systems of retroviral gene regulation that operate through interaction with host transcription factors might specifically target molecular components of the immune response, facilitating their co-option in the immune system specifically.

### HERVs and autoimmune disease

Autoimmune diseases (AD) are complex pathologies defined by a breakdown of immunological tolerance to self-molecules, and the resulting immune responses mounted against those components of the body (*Smith & Germolec, 1999*). However, some of these aberrant responses (and their associated diseases) are not well-defined physiologically, raising the possibility that some autoimmune diseases are actually multiple phenotypically distinct conditions with similar manifestations. Therefore, in attempting to understand their etiology, it must be noted that some causative mechanisms of disease pathogenesis might not apply to the entire patient population. While genetic background seems to affect incidence rates of AD, it is not sufficient for the development of most forms of autoimmunity (*Meda et al., 2010*). To quantify the significance of genetic factors in the development of a given disease, monozygotic twin concordance rates (rates of disease co-incidence) can be measured. Incomplete concordance is observed in many autoimmune diseases (*Bogdanos et al., 2012*), implicating a mix of genetic and environmental factors in their etiology.

The etiology of multiple sclerosis (MS), for example, appears to be associated with components on multiple levels. Multiple sclerosis is an autoimmune disease that affects millions of people, characterized by inflammation of the myelin sheaths that coat nerve axons (*Hauser & Oksenberg, 2006*). Early studies on twin concordance rates in a North American population revealed a partial genetic association for the disease (*Islam et al., 2006*). Using a different approach, genome-wide association studies have identified SNPs that correlate with MS incidence (*Alcina et al., 2011*). Epigenetics are also thought to be involved in the development of MS (*Küçükali et al., 2015*), yet the specific mechanisms by which they contribute remain unclear. A number of environmental factors have been identified to correlate with disease development, including vitamin D, smoking, and Epstein-Barr Virus (EBV) (*O'Gorman, Lucas & Taylor, 2012*).

HERVs appear to have a significant connection with certain autoimmune diseases, but much controversy has surrounded their exact relationship, whether as innocuous by-products, indirect contributors, or directly causative agents of disease symptoms (*Balada, Vilardell-Tarrés & Ordi-Ros, 2010*). Regardless of their exact contribution to disease pathogenesis, a large body of evidence regarding HERV association with AD, consisting of multiple unrelated observations in distinct pathologies, has accumulated over the years. Some of the earliest evidence for HERV association with certain AD

emerged through observations of increased HERV RNA expression through techniques like RT-PCR (*Garson et al., 1998*; *Ogasawara et al., 2001*). Other work has identified anti-HERV antibodies in the blood of patients afflicted with rheumatic autoimmune disease (*Hervé et al., 2001*). More recently, HERV protein production has been compared in healthy individuals and MS patients, revealing certain HERV products that appear to be overexpressed in disease (*Laska et al., 2012*). However, despite extensive evidence of correlation, limited progress has been made in determining the specific contributions of HERVs to the induction of autoimmunity.

### Rheumatoid arthritis

Nevertheless, large strides have been made in improving our mechanistic understanding of HERV involvement in certain autoimmune diseases. The most prevalent of these is rheumatoid arthritis (RA), an autoimmune condition that affects approximately 5 in 1,000 people (*Aletaha & Smolen, 2018*). The main pathological feature of RA is inflammation of synovial joints, which is mediated by a variety of immune system cells and molecules (*Guo et al., 2018*). Autoantibodies appear to play a key role in disease progression, but large variation exists in the specific autoantibody epitope profiles observed in the sera of infected patients (*Aletaha & Blüml, 2016*). Our current understanding of HERV involvement in RA is supported by a large body of evidence, and therefore the disease will be used here as an illustrative example, to demonstrate how mechanistic hypotheses of HERV contributions to autoimmune disease are formulated and tested.

A landmark experiment in the early 2000s was the first to establish a strong connection between human endogenous retroviruses and RA incidence (*Hervé et al., 2001*). Hervé and colleagues compared sera from RA patients to healthy controls and demonstrated that a significantly greater proportion of RA patients were seropositive for antibodies specific to envelope proteins from two members of the HERV-K family: HERV-K10 and $IDDMK_{1,2}22$. Importantly, their group also mapped the epitopes of anti-HERV-K antibodies extracted from diseased patients, and identified multiple distinct reactive peptide epitopes, all arising from HERV-K sequences. These results implied that the antibodies detected by their group were, in fact, driven by a specific response to HERV-K antigens. Later work used RT-qPCR to more precisely quantify expression of HERV-K10 *gag* mRNA in peripheral blood mononuclear cells (PBMCs) extracted from RA patients, revealing significantly higher expression levels in rheumatoid patients compared with both disease and healthy controls (*Ejtehadi et al., 2006*). The group also noted that HERV-K10 *gag* expression was increased almost two-fold at sites of active inflammation, as opposed to the peripheral blood.

More evidence of HERV involvement was presented in a 2010 study that used bioinformatics analyses to map sequence alignments between HERV and human autoantigens, in order to identify potential cross-reactive epitopes (*Freimanis et al., 2010*). Their findings revealed multiple pairs of HERV and host proteins with shared epitopes, including one specifically cross-reactive pair in HERV-K10 Gag1 protein and human type II collagen. Type II collagen autoantibodies were known to be associated with RA (*Choi et al., 1988*), and so Freimanis and colleagues hypothesized that anti-HERV-K Gag1 antibodies, which had previously been observed in RA patients (*Hervé et al., 2001*), could

be cross-reacting with human collagen *in vivo* and contributing to disease pathogenesis. Crucially, this hypothesis predicted that their group could expect to find, in the sera of RA patients, autoantibodies cognate to the specific HERV-K10 Gag1 epitope identified in their bioinformatics analysis. Indeed, they performed ELISA assays to quantify the levels of antibodies specific to this epitope present in the sera of RA patients, and observed titers significantly higher than all controls, including multiple non-RA disease control groups. Taken together with previous evidence of HERV-K antigen-driven autoantibody production (*Hervé et al., 2001*) and increased HERV-K10 *gag* expression levels in RA (*Ejtehadi et al., 2006*), the results obtained by *Freimanis et al. (2010)* suggest a pathological role for HERV-K10 *gag* sequences in disease. Specifically, their findings indicate that an autoantibody response against a HERV-K Gag1 epitope that cross-reacts with type II collagen may contribute to certain manifestations of RA, targeting immune system effector functions to synovial joints.

Subsequent findings affirmed the correlation between anti-HERV-K Gag1 antibodies and rheumatoid arthritis, but failed to identify a correlation between serum antibody concentrations and disease severity (*Nelson et al., 2014*). However, it should be noted that Nelson's group used a Gag1 epitope different to the type II collagen cross-reactive sequence identified by *Freimanis et al. (2010)*. There is ongoing debate about which HERV-K sequences are the most likely to induce the production of cross-reactive antibodies in RA. Thus, while some evidence supports a pathological role for HERVs in disease, such theories must be substantiated by observations of specific correlations between HERV-K activity and disease progression. The works highlighted in this section are not used to present an argument that HERVs act as etiological agents in RA. Rather, they provide a subject-specific example to illustrate how predictive, mechanistic hypotheses in investigations of disease etiology can be formulated from existing evidence. In the remainder of this review, I use the available evidence to construct novel hypotheses regarding HERV involvement in two diseases that are highly-associated with HERVs, but less well-understood: Multiple Sclerosis and Lupus.

### Multiple sclerosis

Multiple sclerosis is a well-studied autoimmune disease that exhibits a strong correlation with expression of multiple HERV families (*Morandi et al., 2017*). Despite the disease's prevalence, and substantial evidence of correlations between HERV expression and disease incidence, our mechanistic understanding of HERV involvement in MS has lagged behind our understanding of their involvement in RA. Determining the underlying reasons for the increased HERV expression observed in MS could improve our understanding of their connection with the disease, and with autoimmunity in general. EBV is one interesting factor to consider. The virus has not only been detected in the sera of 100% of MS patients (*Pakpoor et al., 2013*); it also appears to activate expression of certain HERV elements in both healthy and MS cell lines (*Sutkowski et al., 2001*; *Mameli et al., 2012*). The tripartite relationship between HERVs, EBV, and MS has received some attention in immunological research, and there is substantial evidence of EBV's ability to induce increased HERV expression in cells from MS patients, both *in vitro* (*Mameli et al., 2012*)

and *in vivo* (*Mameli et al., 2013*). Other viruses like HIV and Herpes Simplex Virus have also been demonstrated to interact with endogenous retroviral sequences (*Woo et al., 2003*; *Bhardwaj et al., 2014*). It is well-established that infections can contribute to the development of various autoimmune diseases (*Ercolini & Miller, 2009*), and these findings may provide a link between autoimmunity triggered by exogenous viral infection, and the increased HERV expression observed in many AD. In fact, in light of strong evidence that some HERV-derived promoter regions are regulated by immune system signalling molecules (*Manghera & Douville, 2013*; *Chuong, Elde & Feschotte, 2016*), it is possible that transcription factors like STAT1 –whether induced by infection, or another factor –could be sufficient to activate certain HERV regulatory elements and their downstream genes.

Using a different approach to investigate HERVs in MS, genome-wide association studies have recently attempted to characterize distinct HERV variants that correlate with disease. It was discovered that certain SNPs in sequences of the HERV-K family exhibit significant correlations with MS incidence (*Nexø, 2018*). Some of the risk alleles identified by Some of the risk alleles identified by Nexø and colleagues were SNPs located upstream of their adjacent HERV-K genes, suggesting that certain promoter alleles might induce increased expression of their downstream viral elements in MS patients. These findings provide strong evidence of HERV-K contribution to MS, but genome-wide association studies are not always suitable for identifying complex, multifactorial mechanisms of disease induction. Autoimmune diseases are particularly problematic, being defined in most cases by organism-level immune system interactions that could possibly be induced by multiple distinct molecular mechanisms. Even assuming a strong genetic basis for disease, specific variants at multiple distinct HERV loci might be capable of producing identical pathological effects, further reducing association between any individual gene and disease incidence. The extent of this problem is realized when one considers the vast number of ERV sequences in the human genome and the fact that many of them exist at large copy numbers with high levels of sequence similarity, each group having arisen from replication of a single endogenized retrovirus (*Belshaw et al., 2005*). Therefore, understanding the etiology of autoimmune diseases (especially with regards to HERVs) requires the formulation of specific, testable, and mechanistic hypotheses of pathogenesis. Such hypotheses can be tested in animal models with genetic knockouts or modifications of proposed components, so that disease manifestations can be evaluated on a whole-organism level and theories can be rigorously evaluated. Here, I present one potential mechanism through which HERV elements might contribute to the manifestation of MS. Many other theories have also been proposed (*Perron et al., 2001*; *Balada, Ordi-Ros & Vilardell-Tarrés, 2009*; *Hummel et al., 2015*; *Nexø, 2018*), and it should be noted that most of them are not mutually exclusive.

Many HERV proteins exhibit sequence homology to proteins produced by exogenous retroviruses. As has been demonstrated in mice, this structural similarity can allow ERV-encoded products to serve as antiviral restriction factors (*Pincus, Hartley & Rowe, 1971*; *Limjoco et al., 1993*) . However, HERV molecular mimicry might also be capable triggering antiviral responses in the absence of exogenous viral infection. There is evidence to suggest that HERV-encoded products could contribute to the activation of dendritic cells (DCs) in MS, an important factor in the initiation of disease that leads to various

downstream effects that produce inflammatory symptoms (*Grigoriadis & Pesch, 2015*). One possible mediator for this interaction is TRIM5α. As previously mentioned, TRIM5α is a retroviral restriction factor that inhibits viral replication through interaction with the viral capsid (*Sastri & Campbell, 2011*). In addition to its direct antiviral activity, TRIM5α also contributes to innate immune signalling in dendritic cells, through K63-linked ubiquitination and activation of the TAK1 kinase complex, which induces activation of the transcription factors NF-κB and AP-1 (*Pertel et al., 2011*). Pertel and colleagues showed that the effect of TRIM5 α-mediated signalling was highly specific, with TRIM5 α knockdown only significantly decreasing expression of 33 genes, about 70% of which encoded pro-inflammatory cytokines. Crucially, they also demonstrated that production of inflammatory cytokines in DCs was upregulated in response to increased TRIM5 α interaction with retroviral capsid lattices. There is evidence that miscalibration of these signalling pathways could play a role in the induction of MS.

Constitutive TRIM5 α activity provides a plausible explanation for the aberrant DC activation observed in MS. One large genome-wide association study with over 800 participants revealed correlations between multiple SNPs in TRIM5 α and MS (*Nexøet al., 2013*). The significant SNPs identified by Nexøand colleagues were located at the 5′ end of the gene, near TRIM5 α's RING finger domain. The RING domain has been shown to be responsible for the protein's self-ubiquitination activity (*Keiko et al., 2008*). Keiko and colleagues demonstrated through proteasome inhibition that ubiquitination of TRIM5α does not target it for degradation; instead, it was shown to affect protein localisation, causing TRIM5α to diffuse from localized cytoplasmic bodies into the cytoplasm. It was later demonstrated that the RING domain of human TRIM5α also mediates the protein's K63 ubiquitination activity on TAK1 kinase; TAK1 ubiquitination results in activation and induction of AP-1 and NF-κB in DCs (*Pertel et al., 2011*). The RING domain thus appears to determine TRIM5α's inherent signalling capabilities, as well as regulating the extent to which it interacts with viral capsids (as a consequence of protein localization). As was previously described, TRIM5α-mediated downstream signalling has been shown to be induced by interaction with retroviral capsids, so protein localization could indirectly affect the protein's signal transduction properties (*Pertel et al., 2011*). Therefore, the MS-associated SNP differences in the RING domain of TRIM5α (*Nexøet al., 2013*) could alter the protein's signalling activity in two ways; either by increasing its TAK1 ubiquitination activity or by affecting its self-ubiquitination and localization, thus altering the frequency of protein-capsid interactions. Overall, the current evidence suggests a potential mechanistic link between TRIM5α polymorphism, interaction with retroviral capsids, and dendritic cell activation.

We have seen that Fv-4, a protein encoded by an ERV *env* gene in mice, maintains high levels of structural similarity to its counterpart in exogenous MuLV, and that this similarity allows the endogenous retroviral element to interact with host proteins that recognize exogenous retroviruses. A similar interaction could be occurring between HERV Gag proteins and TRIM5 α. Due to their sequence homology (*Mueller-Lantzsch et al., 1993*), some HERV capsid elements (*gag*) likely overlap with exogenous retroviruses in TRIM5α binding specificity. In addition to TRIM5α sequences, recent findings have

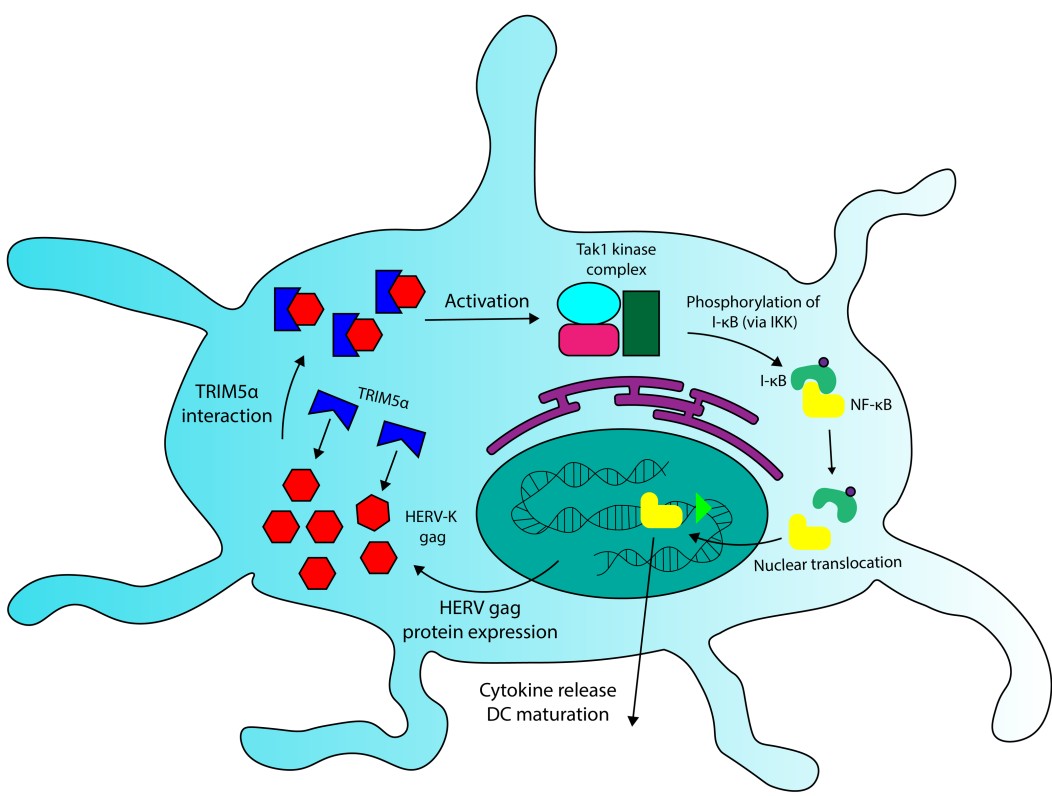

**Figure 4  A role for TRIM5a in dendritic cell activation in multiple sclerosis.** HERV Gag proteins, especially those that are overexpressed in MS patients, could contribute to DC activation by interacting with TRIM5a. TRIM5a interaction with retroviral capsid proteins has been demonstrated to induce NF-κB activation and cytokine production in DCs.

revealed that certain HERV-K SNPs also correlate with MS incidence (*Nexø et al., 2015*), and that expression of some HERV-K *gag* genes is significantly increased in PBMCs and brain cells extracted from MS patients (*Laska et al., 2012*; *Bhetariya, Kriesel & Fischer, 2017*). These HERV-K variants could comprise overactive regulatory sequences (LTRs) or *gag* proteins with increased binding affinity for TRIM5α, though the observations of increased HERV-K protein expression in MS (*Laska et al., 2012*; *Bhetariya, Kriesel & Fischer, 2017*) provide evidence for the relevance of the former. Because TRIM5α interaction with viral capsids increases the protein's downstream signalling activity in DCs (*Pertel et al., 2011*), HERV *gag* variants with high avidity for the restriction factor, or variants that are overexpressed, have the potential to increase inflammatory cytokine production through TRIM5α-Gag interactions. I posit that the aberrant activation of DCs in MS may arise due to the combination of TRIM5α variants with increased signal transduction properties and HERV-K variants that increase TRIM5α-mediated signalling. Thus, the combination of rare SNPs at HERV and/or TRIM5α loci could provide a genetic basis for Multiple Sclerosis. A detailed mechanism of HERV and TRIM5α-mediated activation of DC cells is shown in Fig. 4.
To support this hypothesis, a more comprehensive genomic analysis is required to determine to covariance of high-risk alleles at HERV-K and TRIM5α loci in the incidence of MS. Future research should also investigate the binding specificity of TRIM5α for HERV-encoded Gag proteins, and whether this interaction has any downstream effects in immune system cells, as occurs with exogenous retroviral capsids. Despite the current lack of information, TRIM5α and HERV-K Gag interactions present an exciting new avenue for future investigation into the molecular basis of MS pathogenesis.

### Systemic Lupus Erythematosus

Systemic lupus erythematosus (SLE) is another autoimmune disease that appears to have a significant relationship with HERVs. HERV association with the disease has been established through observations of increased HERV mRNA expression in afflicted patients, as well as the detection of circulating antibodies specific to HERV elements (*Ogasawara et al., 2001*; *Hervé et al., 2001*). SLE is a chronic inflammatory autoimmune disease, characterized by an abnormal interferon response that causes an individual to produce auto-reactive antibodies and immune complexes that target the body's own tissues and organs (*Maidhof & Hilas, 2012*). One of the first pathological features of the disease to be detected was the presence of anti-dsDNA antibodies in the sera of affected patients; antibody concentration was observed to correlate with disease progression (*Swaak et al., 1982*). Type 1 activation of macrophages (a pro-inflammatory response) also appears to play a key role in disease manifestation, as non-aberrant macrophage activation induces SLE remission in mice (*Schiffer et al., 2008*; *Li et al., 2015*).

The factors that induce pro-inflammatory activation of macrophages in SLE are not yet fully understood, but dysregulation of the previously-mentioned AIM2 protein complex provides a feasible mechanism. AIM2 is a foreign nucleic acid sensor, part of a group of proteins that recognize so-called pathogen-associated molecular patterns (PAMPs) (*Roers, Hiller & Hornung, 2016*). The AIM2 complex recognizes cytosolic DNA in a sequence-independent manner (*Jin et al., 2012*), and upon recognition initiates an inflammatory response involving the maturation and release of pro-inflammatory cytokines (*Di Micco et al., 2016*). As macrophages commonly phagocytose microbes and other pathogens, this DNA-sensing mechanism allows the cell to attune itself to an active infection. Yet as well as pathogens, macrophages also engulf and digest apoptotic host cells (*Leers et al., 2002*), and AIM2's sequence-independent recognition of DNA likely allows it to activate upon binding apoptotic DNA. This has been observed in macrophages extracted from SLE patients, in an experiment that also noted correlations between AIM2 expression and SLE disease severity in humans (*Zhang et al., 2013*). Zhang and colleagues also designed an SLE murine model, in which AIM2 knockdown significantly alleviated disease pathologies in afflicted mice. Even more interestingly, their experiment showed that increased AIM2 expression in the mouse model correlated with increased anti-dsDNA antibody levels in serum, a feature associated with human SLE pathogenesis (*Swaak et al., 1982*; *Živković et al., 2014*). AIM2 has separately been implicated in the induction of MHC class II in cancer cells (*Lee et al., 2011*), an antigen presentation complex expressed in macrophages (*Harding & Geuze, 1992*). Aberrant AIM2 activity might therefore contribute to the production of anti-dsDNA

antibodies observed in SLE patients, through activation antigen presentation via MHC class II and induction of a CD4+ T cell response against dsDNA antigens (*Waisman, Zisman & Mozes, 1996*; *Underhill et al., 1999*). Regardless of its role in autoantibody formation, however, the evidence accrued by Zhang and colleagues (*2013*) indicates an indispensable role for AIM2 in the pathogenesis of SLE.

As previously described, AIM2 operates (in part) under the control of *MER41.AIM2*, a single HERV-derived promoter region. There is evidence that the etiology of SLE might relate to epigenetic dysregulation of *AIM2*. A high-throughput analysis of CpG-containing promoters for various genes revealed significant differences in *AIM2* promoter methylation ($p = 0.01$) in monozygotic twin pairs disconcordant for SLE (*Javierre et al., 2009*). Individuals with SLE exhibited hypomethylation at the locus and corresponding increased expression of AIM2 compared with their healthy twin. While the analysis did not investigate the specific nature of each promoter tested, *MER41.AIM2* is known to be located in very close proximity to *AIM2* and has been demonstrated to be a STAT1 binding site, necessary for *AIM2* expression under IFN-γ stimulation (*Chuong, Elde & Feschotte, 2016*). It therefore serves as a probable candidate for epigenetic modifications that affect *AIM2* expression. Given that *AIM2*'s activity appears to be intimately linked with SLE pathogenesis (*Zhang et al., 2013*), the hypomethylation identified at the gene's promoter by *Javierre et al. (2009)* provides a plausible mechanism for *AIM2* overexpression and suggests that *MER41.AIM2* dysregulation might contribute to SLE pathogenesis, at least in some patients. There remains much to be understood about the prevalence of co-opted ERV regulatory sequences in the human genome and immune system, but given our current knowledge of ERV co-option in other vertebrates, investigation into such elements presents promising opportunities for future research. A more comprehensive analysis with larger sample sizes and more thorough epigenetic profiling is necessary to elucidate the exact modifications that contribute to SLE. Further research into the dysregulation of HERV promoter regions in SLE and other autoimmune diseases could reveal additional associations with disease incidence, given their intimate relationship with the immune system.

The previously-detailed evidence suggests a meaningful role for AIM2 in SLE pathogenesis, but Toll-like receptor 7 (TLR7)—another nucleic acid-sensing system—might also contribute to disease progression through the detection of HERV-encoded products. TLR7 is an endosomal ssRNA-binding receptor—expressed in immune system cells—that triggers a pro-inflammatory response upon binding an ssRNA ligand (*Brubaker et al., 2015*). SLE patients exhibit increased expression of TLR7 in peripheral blood mononuclear cells (PBMCs) (*Guo et al., 2015*). Guo and colleagues also demonstrated that PBMCs from these patients were more sensitive to TLR7 stimulation by ssRNA. TLR7 has also been shown to be capable of recognizing ERV-derived RNA in mice (*Yu et al., 2012*). ERVs have been implicated in the pathogenesis of murine SLE, as early investigations revealed that the ERV-encoded Env protein gp70 was present in the sera of multiple lupus-prone and healthy mouse strains, but that only sera from SLE-prone mice contained anti-gp70 immune complexes (*Izui et al., 1979*). Mutant SLE mice with a TLR7 duplication exhibit increased levels of anti-ribonucleoprotein antibodies, including

anti-gp70 complexes (*Santiago-Raber et al., 2010*). Santiago-Raber and colleagues also demonstrated that B cell treatment with a TLR7 agonist significantly increased B cell activation in their murine model. Other work has demonstrated that RNA-associated autoantigens activate autoantibody production in SLE mouse B cells via combined activation of BCR and TLR7 (*Lau et al., 2005*). PBMCs in humans with SLE have been shown to overexpress TLR7 (*Guo et al., 2015*), which might increase the production of anti-RNA autoantibodies, as it does in mice (*Santiago-Raber et al., 2010*). If TLR7 overstimulation does play a role in SLE pathogenesis, the effect is likely exacerbated by the increased concentrations of HERV mRNA—a TLR7 ligand—that are associated with disease (*Ogasawara et al., 2001*).

Being derived from exogenous retroviruses, whose RNA and protein elements spontaneously assemble, certain ERV elements may have the potential to produce complexes consisting of immunogenic viral protein antigens associated with TLR7-stimulatory ssRNA. Such complexes could be sufficient for co-stimulating BCR and TLR7 in B cells, even if the viral particles themselves are not infectious. A model for this interaction is displayed in Fig. 5. HERV virion-RNA complexes have been detected in the sera of MS patients (*Garson et al., 1998*). Further supporting this theory, anti-HERV-K Env antibodies have been detected in the sera of human SLE patients (*Hervé et al., 2001*). However, in order to confirm the relevance of TLR7 and HERV interactions in SLE, a more robust connection must first be established between specifc molecular components and disease incidence. Genome-wide association studies could help to identify TLR7 or HERV variantsthat are associated with disease. Epitopes from anti-HERV-K Env antibodies in SLE patients could also be mapped to specific loci in the genome, which would allow for more robust comparison between SLE-associated and healthy alleles at HERV-K sequences (assuming such variants exist). As was discovered in rheumatoid arthritis (*Freimanis et al., 2010*), some of these epitopes might be cross-reactive with human proteins. *In vitro* experiments could also test the ability of HERV virion-RNA complexes to stimulate TLR7 in B cells, assuming such complexes could be isolated or produced *in situ* . If a connection is established between TLR7/HERV interaction and disease pathogenesis, the receptor presents a suitable target for treatment via inhibition, due its functional redundancy with TLR8 (*Yang et al., 2005*)

## CONCLUSIONS

The relationship between HERVs and autoimmune diseases has long been a controversial topic, and a comprehensive model for their involvement in autoimmunity has not been synthesized. Autoimmune diseases are particularly complex pathologies, usually not defined by specific alleles or pathogens, but instead by the symptoms they produce and the complex interactions observed in the immune systems of diseased patients. Therefore, it might be inaccurate to view them as single diseases, but rather as sets of systemic effects whose contributing factors correlate frequently. This uncertainty makes it difficult to make meaningful statements about AD etiology; a diversity of factors could be capable of independently producing the same condition. The purpose of this review is to summarize

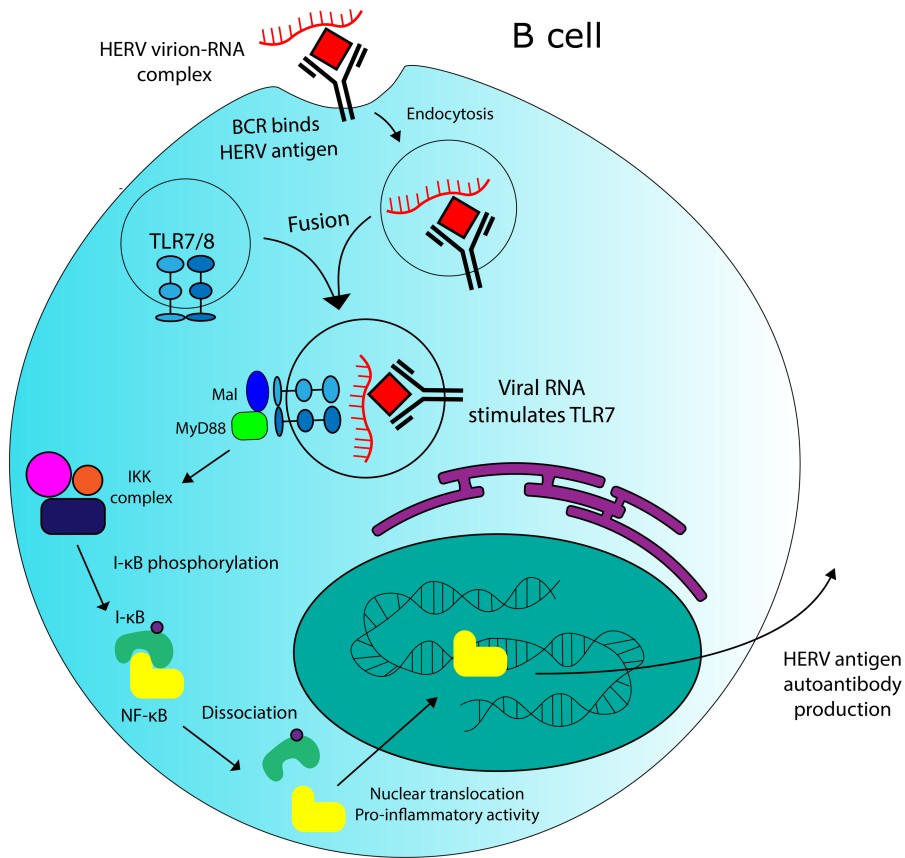

**Figure 5** **Potential role for HERVs in autoantibody production in SLE.** There is evidence that HERV RNA-protein complexes formed in SLE patients could be capable of stimulating production of autoantibodies against HERV antigens. TLR7 stimulation by HERV RNA could contribute to this effect.

some of our knowledge with regards to HERV involvement in immunity and autoimmunity, and to present some novel theories concerning the mechanistic involvement of HERVs in autoimmune pathologies.

Co-opted ERV elements have been identified in animals and humans, but our current understanding of human genomics cannot provide an accurate estimation of their biological significance. ERV-encoded restriction factors have been identified in mice and other animals, but human ERVs remain far less well-characterized. It has been observed that HERV activity correlates with the incidence of multiple autoimmune diseases in humans, but specific mechanistic hypotheses are required to determine their exact relationship. In this review, I argue that HERV elements are not only capable of serving beneficial functions in their hosts; they may also have to potential to trigger pathological immune responses by producing immunostimulatory viral antigens or acting as uncontrolled regulatory sequences. The domains of paleovirology and immunology will continue to overlap as more is discovered about present-day HERV activity in the immune system. Investigation of endogenous viral elements is not only relevant to evolutionary biology; it may also shed light on the nature of such autoimmune pathologies as lupus and multiple sclerosis.

## ACKNOWLEDGEMENTS

I would like to thank Dr. Michael Tristem for serving as my primary supervisor in the production of this review. His insight and advice have been invaluable, and the project in its current form would not exist without him.

### Funding

The authors received no funding for this work.

### Competing Interests

The authors declare they have no competing interests.

### Author Contributions

- Matthew Greenig prepared figures and/or tables, authored or reviewed drafts of the paper, approved the final draft.

### Data Availability

The research in this article did not generate any data or code. The article is a literature review that references research published in scientific journals online, but does not include any of its own research.

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

## FURTHER READING

**Emerman M, Malik HS. 2010.** Paleovirology—modern consequences of ancient viruses. *PLOS Biology* **8(2)**:e1000301 DOI 10.1371/journal.pbio.1000301.