# Peer review of "HERVs, immunity, and autoimmunity: understanding the connection"

_PeerJ, doi:10.7717/peerj.6711_

## Round 0.1 · original submission · Major Revisions

The manuscript has been reviewed by three experts in the filed, and some important questions were raised, especially regarding to the lack of relevant information. This is pointed by the referees in their specific comments and the authors need to complement the text accordingly. It is also advisable that the authors should enrich the text with some examples of specific human diseases and the possible role of ERVs in these as marked by the reviewers.

·

Basic reporting

In this manuscript, Greenig reviews and synthesizes recent findings relating to endogenous retroviruses in immunity and autoimmunity. Overall the review is very well written and was a pleasure to read, being both informative and insightful. The figures are also original and extremely well done. While there have been a few reviews relating to HERVs and immunity (in an evolutionary sense) or autoimmunity (from a disease-causing sense), no other review has directly bridged these largely distinct subfields. The unique cross-disciplinary perspective is properly introduced. Overall this is an excellent review which should be of interest to both specialists and newcomers to the topic of endogenous retroviruses and immunity.

Experimental design

The author first introduces the concept of ERVs. Then, he summarizes key findings of ERVs co-opted for immune responses for both protein-derived and regulatory functions. Key cases include mouse Fv1 and Fv4, and human MER41. In the next section, he summarizes our current understanding of HERVs in autoimmune diseases, focusing on multiple sclerosis and lupus as examples. The author next proposes several mechanisms linking co-opted ERVs to disease-causing ERVs. First, he notes that AIM2 is regulated by an ERV, and aberrant AIM2 expression is linked to autoimmune diseases including lupus. Therefore, epigenetic dysregulation of a co-opted ERV promoter may contribute to disease. In a second example, he discusses HERV mRNA as a potential source of TLR7 overstimulation in lupus. In a final third example, the author speculates that polymorphism of HERV-K alleles may have modifier effects on Trim5a activity, which may contribute to multiple sclerosis.

Major comments:
-While several key studies are reviewed as illustrative examples, it would be useful to add a comprehensive table (or two) summarizing all published/experimentally validated examples where ERVs are contributing to immune function or disease with references, species, level of evidence, etc.
-Another example of co-option could be listed, ERV-derived dsRNA can also contribute to immune “priming” (Canadas et al, Nature Medicine 2018) which may connect with the proposed mechanism with TLR7.
-More disease-related examples could be drawn from neurodegenerative diseases, including ALS and schizophrena, which have some mechanistic overlap with autoimmune diseases.
-The sections (“HERVs and autoimmune disease” and “HERVs and autoimmune pathogenesis”) could be distinguished more as separate sections (or simply combined). The current organization is currently a bit confusing jumping back and forth across autoimmune diseases. Perhaps the first could specify “Current evidence associating HERVs and autoimmunity” and the second may be “Potential mechanisms linking HERVs to autoimmunity”


Minor comments

Line 196: Not necessary to report the P value from another study
Line 330: This paragraph (HERVs and SLE) was difficult to follow and contained many details that were not relevant from a clear argument. It should be rewritten for clarity.

Validity of the findings

The conclusions of the review are well-stated and provides insightful outlook relating to the challenges of linking HERV activity to autoimmune disease. I would suggest adding some additional discussion on how our understanding of HERV contributions to disease may impact the use/interpretation of animal models in autoimmune research.

Additional comments

This is a very nice and thoughtful single-author review, and very impressive considering the career stage of the author (undergraduate).

Reviewer 2 ·

Basic reporting

In this manuscript, the author reviewed literature about endogenous retroviruses with an emphasis on their impact on the host immune system, which fits in the general scope of PeerJ. However, some fundamental issues needs to be addressed to make it suitable for publication, so we recommend a major revision.

The scope of the review needs to be better defined. The manuscript is a update of recent advances in immunity-related topics of endogenous retrovirus. This pertains to two major areas: ERV co-option into the host immune system, and ERV-related autoimmune disease, both of which have been reviewed recently. Although some recent reviews have been cited, the author needs to layout a clear scope of what have been covered in those recent reviews and what have not been, then clearly define the scope of this review and focus on it.

The structure of the writing borders on superficial treatment of an extremely broad range of topics and points. The author should define the important points to address the above designed scope and build a clear organization of the text body. One useful revision would be to add a number of subheading to orient the reader and structure the logic.

The body of the manuscript equally addresses ERV co-option and ERV-related autoimmune disease, whereas the introduction almost exclusively talks about the former. We recommend condensing the ERV co-option section of the introduction, and extracting some of the points of the ERV-related autoimmune disease section and move it to the introduction, if not re-writing the whole introduction.

Experimental design

When citing previous literature, the author tends to talk about the finding/conclusions in them. Instead, the author should talk about the evidence found in the former studies, and then synthesize the evidence to support/reject certain points. Some of these cases are listed in the "general comments" section of the review.

Key points need to be emphasized to improve the manuscript. The author listed findings of previous works in the field, but these findings are disconnected given how they are written. As suggested above, a clear definition of the scope of the review will help the author finding these key points, and other parts of the manuscript should revolve around the key points. Such key points could be but not limited to: Why is ERV co-option a unique way to gain immunity, and what are the possible consequences (beneficial or deleterious) to the ERV and the host? Why are ERV-related autoimmune disease interesting compared to those triggered by other pathogens?

Some highly related topics have only been superficially mentioned without further discussion. One of them is the ERV-host arms-race: the impact of ERV on host genome, particularly the evolution of the host innate immune system, is very relevant given the title of the manuscript. The other case is rheumatoid arthritis in ERV-related autoimmune disease. There are rich literature about ERVs in rheumatoid arthritis, but it was only mentioned once in the manuscript without further discussion. This is very surprising given the detailed review of ERVs in MS and SLE in the manuscript.

The figures are essentially adaptations of textbook or other article figures without synthesis to support the focus of this review (when defined). A summary figure that shows how ERVs can affect the immune system will be very helpful to introduce the readers to the topic. Figure 1 can potentially be adapted to such a figure with elements added. Indeed, although figure 1 shows the life cycle of retrovirus, it probably isn't worth a figure on its own.

Validity of the findings

no comment / see above and below

Additional comments

Title: the author needs to be careful on where to use "ERV" and "HERV". Although the manuscript is titled to talk about "HERVs", some of the points discussed in the paper are not specific to humans. In fact, the co-option stories are all based on rodent genomes. Also, in the introduction, only use "HERV" when a certain property of an ERV is specific to human or only human-based evidence is available.

Figure 2: there is significant evidence that ERV transcripts are also important regulators of immunity (Wysocka, McFarland, Coyne, others).

Multiple lines: "et. al" should be "et al." and italic.

Line 71-72: retroviral RT happens in the cytoplasm (fig1 is correct).

fig1 do all retroviruses enter via endocytosis, not membrane fusion?

78-80: confusing presentation. Are you talking about new ERVs? Integration could be in gamete, germline, or early embryo.

80: "now" should be changed to "then".

82-85: humans are exceptional, as there are tons of species with active ERVs

89: "most significant" not appropriate

90: syncytins, also expressed in placenta not just precursor

94: syncytins are not HERV domestications

98: some intact ERV ORFs are not old enough to expect them to decay, decay is probabilistic

100: tissue breadth, not proportion of transcriptome

103: 'biological role' unclear meaning

110: intuition suggests (Delete)

115-126: "propositions are novel…fair and accurate…" This whole section is inappropriate; these are judgments of reviewers. We suggest rewriting or removing it.

147: ERV not HERV, also see comments on the title

165 and fig3: there is no mention of 'rough ER', etc in text. Citation needed.

212-213: the connection here needs to be clarified.

215: need a definition of an autoimmune disease.

219: “MZ” only appeared once in the text and do not need to be abbreviated. We suggest going through the manuscript and see if there are additional cases like this.

233-234: EBV-HERVs? This connection appears abruptly and only later discussed.

236-243: there is a lot of statements without the important part which is the data. This section jumps around a lot - MS, EBV, others, MS, EBV…

252, 253: why is nexo underlined?

291: need a reference

303-305: this could probably be said for many immune genes.

402: "gag" should be italic. We suggest doing a global search in the manuscript for latin-derived nomenclatures. Also, "Env" should be "env" and italicized in multiple locations of the manuscript.

Reviewer 3 ·

Basic reporting

This is a review on a very interesting topic, human endogenous retroviruses and their role in immunity and autoimmune disease. My overall feeling is that the review is not sufficiently broad to cover the title. The review focuses on very specific aspects of HERVs in relationship to immunity and autoimmunity and fails to cover all relevant aspects. Having said that i feel that if the review title was more specific, then it could make a sensible contribution, although I think the authors need to find how exactly this is going to happen. Some examples of missing important aspects (not an exhaustive list) relevant to the topic are info on the immunosuppressive properties of retroviral envelopes, the role of HERVs in relevant diseases (Aicardi Goutieres, ALS) etc.

Experimental design

See comments above

Validity of the findings

See comments above

Additional comments

-This is a review focusing on human ERVs, thus I am not sure what exactly is the purpose of the fV-4 in mice and why exactly the fV-4 example and not another ERV?
-ERVs are not only transmitted vertically
-"This hypothesis align with the observation that nearly all human ERVs....": I think this is not exactly true. There are active ERVs in animals which are also hosts thus if the above statement was the explanation of ERV inactivation this should be true for all hosts.
-"...pertaining to present day HERV activity...": I am not sure that "present day HERV activity" is a good choice of words for the specific situation.
-"Epigenetics are indisputably": are we so sure about that?
-Please check if the use of the term "homology" is correct where it is used. I think similarity would be more accurate. Homology means common ancestry, thus it is a "black-white" term. Two genes are either homologous or not.
-The acknowledgement of the manuscript suggests that Dr Mike Tristem has made a significant contribution, however he is not involved in the authorship? According to authorship criteria he could probably be a co-author. Has he been offered the opportunity to be a co-author (i.e. read the manuscript and comment etc) according to the authorship criteria recommended by ICMJE?

---

## Round 0.2 · accepted · Accept

The manuscript gives a full and comprehensible review about the interrelationships between HERVs and immunity.

# ·

Basic reporting

I have read the revised manuscript which has been significantly improved in its clarity and organization. I feel it is now suitable for publication in PeerJ.

Experimental design

NA

Validity of the findings

NA

Additional comments

Great job!

Reviewer 2 ·

Basic reporting

no comment

Experimental design

no comment

Validity of the findings

no comment

Additional comments

In this revision, the author nicely addresses our previous concerns. The manuscript is now clearly defined in scope, focusing on the interaction between ERVs and the host immune system, with particular focus on HERVs and examples of human autoimmune diseases. Compared to the last version, citations are now more concept-oriented instead of data-specific. The introduction of EVEs, then gradually moving to ERVs and HERVs made the introduction more logically sound. The added section pertaining to rheumatoid arthritis not only adds an example, but also highlights the potential role of HERV gag in autoimmune disease.

Overall, we believe that the revised manuscript is suitable for publication but would like to point out the following minor issues:

1) It would be worthwhile to revise the titles and/or figure legends of figures 2-4 to highlight the general principle addressed with the specific example drawn out in the figure. For example, Figure 2 is co-option against the receptor of an exogenous retrovirus.

2) The “host cell” in Figure 1 should be “host germline cell” for the ERV colonization and then co-option to make sense.

We also add that we very much appreciate the constructive tone and approach of the rebuttal letter.

Best,

Rick McLaughlin and Lei Yang